# Application of Laser-Induced Breakdown Spectroscopy in the Quantitative Analysis of Elements—K, Na, Ca, and Mg in Liquid Solutions

**DOI:** 10.3390/ma15103736

**Published:** 2022-05-23

**Authors:** Wojciech Skrzeczanowski, Maria Długaszek

**Affiliations:** Institute of Optoelectronics, Military University of Technology, ul. gen. Sylwestra Kaliskiego 2, 00-908 Warsaw, Poland; maria.dlugaszek@wat.edu.pl

**Keywords:** laser-induced breakdown spectroscopy, liquid solution, filter paper, calibration, factorial analysis

## Abstract

Results of laser-induced breakdown spectroscopy measurements of K, Na, Ca, and Mg content in liquid media are discussed in the paper. Calibration results show correct parameters—linearity and R^2^ coefficients of determination at the levels of 0.94–0.99. Obtained regression equations have been used to determine K, Na, Ca, and Mg concentrations in biological samples with known element content. Measurement results showed acceptable, within the expanded standard uncertainty, conformity with their content in the certified materials. Results have been supported by multivariate factorial analysis, which was especially effective for Ca and Mg samples. For these elements, factorial analysis allows the application of the whole spectra to obtain quantitative data on the tested samples, in contrast to a common method based on the selection of a particular spectral line for the calibration.

## 1. Introduction

Sodium (Na) and potassium (K), as well as calcium (Ca) and magnesium (Mg), the two pairs of elements antagonistically but also synergistically interacting with each other, are spread widely in the environment. Many factors affect their content in various samples. Therefore, the authors give different data on the content of Na, K, Ca, and Mg. Examples of the content of these elements in waters and human fluids are given in Table 1 [1,2,3,4,5]. From a disease point of view, related to arterial hypertension prevention, the correct ratio of Na and K in the diet is very important. According to the European Food Safety Authority data, Na intake should be lower than 2000 mg per day, recommended daily K intake—3500 mg, Ca—750 mg, and Mg—350 mg [6]. Regarding Na and K, in addition to food products, we consume them also through tap water, mineral water as well as coffee and tea beverages. Examples of the element’s content are listed in Table 2 [7,8,9,10].

Instrumental methods such as atomic absorption spectrometry (AAS), inductively coupled plasma optical emission spectroscopy (ICP-OES) and inductively coupled plasma mass spectrometry (ICP-MS) are routinely used in the quantitative analysis of elements [11] and are characterized by lower limits of detection (LOD) compared to LIBS.

The LOD values expressed in µg/L according to [12] are the following: (Na)—0.3, 0.5, 0.00001 for subsequent methods, and 3, 1, 0.00004 in the case of K, 1.5, 0.05, 0.00005 for Ca, and 0.15, 0.04, 0.00001 for Mg. To determine much larger amounts of elements, the samples should be properly prepared (e.g., diluted several times) and the appropriate parameters of the instrumental analysis should be selected. Multiple dilutions of the sample may bias the final result of the analysis.

In this paper, we have made efforts to evaluate the possibility of using another method—laser-induced breakdown spectroscopy—to quantify Na, K, Ca, and Mg in liquid solutions.

LIBS [13] is a long-known, relatively inexpensive technique for the quasi non-invasive testing of various samples of different physical state, various origin, chemical composition, and so on. Descriptions of LIBS applications in biological material can be found, for example, in [13,14,15,16]. Well-known LIBS deficiencies are its relatively higher LODs compared to the previously mentioned instrumental methods. This is mainly due to the physical mechanisms involved in the laser–matter interaction. They include fluctuations in laser beam characteristics, which affect the plasma parameters that determine the strength of the LIBS signal. At the same time, in LIBS measurements, it is very important to maintain the constancy of as many experimental parameters as possible. This can improve the repeatability and precision of LIBS. Of the many quantitative LIBS strategies, we used the calibration curves with the CRM method. For liquid solutions, calibration curves are most frequently based on dropping standard solutions onto filtration papers, while, for solids, powdered and pressed pellets/tablets are applied [16,17,18]. Two papers have recently been published on the use of LIBS in the study of animal tissues, in which 3D elemental mapping was used in [19] to study the brains of mice, and in [20], glycerol was used to determine its effect on the LIBS signal from various tissues (pork liver, fat, and muscle tissue). In any type of LIBS experiment, a multivariate statistical analysis is very helpful.

K, Na, Ca, and Mg belong to the group of elements that are often difficult to determine by the routinely used methods in elemental quantitative analysis, such as AAS, ICP, and ICP-MS. This is due to their rather high concentrations in environmental samples and their low detection limits for the mentioned methods. In this work, we have attempted to develop conditions for the rapid analysis of the concentrations of the abovementioned four elements, in biological samples transferred into a water solution, using the LIBS method.

## 2. Materials and Methods

### 2.1. Preparation of Samples

Samples were prepared as described previously [21]. The K, Na, Ca, and Mg solutions (prepared in the Institute of Optoelectronics Military University of Technology, Warsaw, Poland) with a concentration of 10–1000 µg m/L were pipetted onto circles of filter paper (Munktell & Filtrak, Niederschlag, Germany) of 1.6 cm diameter (2 cm^2^). Then, 15 µL of each calibration solution was prepared in deionized water (Cobrabid-Aqua Sp. z o.o. Warsaw, Poland) from Na, K, Ca, and Mg standards for AAS (Fluka Chemie GmbH, Buchs, Switzerland). Next, they were dried for 2 h and then exposed to laser radiation for 24 h. A Hirox KH 6700 digital microscope (Hirox Co. Ltd., Tokyo Japan, Hirox Europe Ltd. Ecully, France) was used to take pictures showing the structure of a filter paper sample after laser irradiation. An example for an Na sample is shown in Figure 1. To verify our calibrations, three reference materials (SRM 1577b—bovine liver, INCT-T/L—tea leaves, and NCS ZC91002—human hair) were used with certified values for all four elements. The weighed samples of materials (approx. 0.25 g) were mineralized in a mixture of acids—HNO_3_ and HClO_4_ (3:1). The residues after mineralization were transferred to measuring flasks with a volume of 25 mL.

In the case of bovine liver, the certified values for Na and K content are 0.234 ± 0.013 wt.% and 0.999 ± 0.007 wt.%, respectively. In tea leaves, the amounts are the following: 24.7 ± 3.2 μg/g and 1.70 ± 0.12 wt.% for Na and K. In human hair, the Na and K content is Na—266 ± 12 μg/g and K—11.8 μg/g. Regarding the Ca and Mg content in the reference materials, the values are the following: 120 ± 7 μg/g (Ca) and 600 ± 15 μg/g (Mg) in the bovine liver, 0.582 ± 0.052 wt.% (Ca) and 0.224 ± 0.017 wt.% (Mg) in tea leaves and 1090±72 μg/g (Ca) and 105 ± 6 μg/g (Mg) in human hair.

### 2.2. Experimental Setup

In the experiment carried out in the normal environmental conditions, we used a Quantel BigSky laser model Brio (Big Sky Laser Technologies, Bozeman, MT, USA, QUANTEL S.A., Les Ulis, France). It generated 56 mJ of 1064 nm radiation in a pulse of 4 ns width. A laser beam of 4 mm diameter was focused on the flat filter paper using a 150 mm lens. The real distance between the lens and the target was set to 145 mm to avoid a random laser spark in the air. Laser parameters were optimized in such a way that no burn-through occurred at the target. Three laser pulses, necessary for obtaining one useful spectrum, provided reasonable reproducibility and low scattering of results. Radiation of the LIBS plasma was registered using an ICCD camera with a Kodak KAF 1001 detector mounted on the ESA 4000 echelle spectrometer (LLA Instruments GmbH & Co KG, Berlin, Germany). The experimental system (Figure 2), the same as the one used in [21], allowed us to obtain the spectral resolution of λ/Δλ~20,000.

Measurement conditions were selected to ensure a good signal-to-noise ratio; namely, the gate width was equal to 1 µs, the gate-to-laser pulse-delay was 500 ns, and the camera gain was set to 3000, while other details of the experiment can be found in [21]. The results presented later in the text are the average values for five spots on one filter sample (three laser shots were delivered in each spot), which improved repeatability and decreased fluctuations in the registered LIBS signal.

### 2.3. Statistical Processing

Raw LIBS spectra were firstly processed by ESA 4000 software (EsaWin, LLA Instruments GmbH & Co KG, Berlin, Germany, version 13.9.0, S/N 5207), and then with Microsoft 365 (Excel) (Microsoft Corporation, Redmond, WA, USA, Microsoft Sp. z o.o., Warsaw Poland, Microsoft^®^Excel for Microsoft 365 MSO, version 2204, compilation 16.0.15128.20210) and finally by statistical packages (Statistica v.10 PL) (StatSoft Inc. Tulsa, OK, USA, StatSoft Polska Sp. Z o.o., Krakow, Poland). The EsaWin transformed spectra to CSV and/or XLS files. Statistical processing was based on factorial analysis (FA), [22]. The FA is a statistical multivariate analytical technique similar to the often used principal components analysis (PCA), [22,23]. Both approaches, the PCA and the FA, are based on the orthogonal, linear conversion of the input LIBS spectra set to new variables, which are not correlated and not always easy to interpret. These new variables are commonly known as components or factors. In the paper, the input FA matrix consisted of over 2 million elements, with the intensities of 40 LIBS spectra fragments in over 54,000 wavelength intervals covering the entire 200–800 nm spectrometer spectral range. The conversion can be presented graphically, and similarities and differences in input LIBS spectra, reflecting the chemical composition, can be easily seen. The evident benefit of the application of the FA or PCA approach is the simple presentation of similarities and differences in input data sets, obviously at the expense of reducing the input variance. It is generally accepted that FA or PCA procedures are performed correctly when 60–70% of the initial variability is included [23]. In our case, for input data—the LIBS spectra—the FA representation carried over 80% of input variability. The best results were obtained for Ca and Mg, while for K and Na, the FA showed only the correctness of the selection of experimental parameters.

## 3. Results and Discussion

In Figure 1, an exemplary microscopic image of an irradiated filter paper sample is shown (a magnified crater after three laser shots is presented). In the experiment, craters of approximately 350 μm in diameter and 200 μm in depth were usually created. The samples described in Section 2.1 were then used for the calibration of the LIBS signal. In the whole experiment, all parameters (experimental geometry, laser and detection characteristics) were kept the same, both for calibration and biological samples. This allowed us to relate the LIBS calibration curves for K, Na, Ca, and Mg with the content of the elements.

Parts of the LIBS spectra in the regions of interest are shown in Figure 3 (all spectra represent 500 μg/mL concentrations of K, Na, Ca, and Mg solutions dropped onto the filter samples). Our analysis is based on the calculation of the intensities of Na I 588.995 nm, K I 769.896 nm, Ca II 393.366 nm, and Mg II 279.552 nm lines.

Calibration curves in the 0–1000 μg/mL range are presented in Appendix A. As seen from the Ca and Mg curves, the lines of Ca II 396 nm and Mg II 280 nm gave a lower slope, which resulted in higher LOD values, and this is why we decided to carry out measurements on the Ca 393 nm and Mg 279 nm lines only. The counts used as measures of the element content were calculated as net values (the area under the line profile minus the contribution of the background).

Although we used a linear approximation for calibration curves, it can be easily seen in Appendix A that the measurement points show a steeper slope for lower content. Such behavior is mainly caused by the nonlinear dependence of the LIBS signal on the analyte content, which can be explained by the stronger reabsorption for larger K, Na, Ca, and Mg content in the prepared samples. This is especially apparent since all selected spectral lines were resonance ones, which can undergo reabsorption quite easily.

As an example, shown in Figure 4, the calibration curve for K, as with those for the other three elements, is not a single linear trend line across the whole content range—it can be better approximated by two linear trend lines, which are particularly justified for low content (0–250 µg/mL), for which the slope is five times higher than for the 250–1000 µg/mL interval.

Such an approach is correct for all four elements since their content in the reference samples is lower than 250 µg/mL. Calibration curves for low element concentrations (0–100 µg/L and 0–250 µg/mL) are shown in Appendix A. Corresponding equations of regression, as well as coefficients of determination for low content, are also shown in Appendix A. The K I 769.9 nm line intensities were much lower than those for the Na I, Ca II, and Mg II lines. This resulted in higher standard deviations of the measured values and lower coefficients of determination. Unfortunately, due to the optical arrangement of the ESA 4000 spectrometer, the more intense 766.5 nm K I line could not be observed, which is shown and explained in Appendix A. In general, the results presented in Appendix A show the correct choice of experimental parameters such as sample size or the procedure of preparing standard samples, as well as the calibration procedure.

After calibration, measurements on real CRMs were performed. We measured the concentrations of the four elements (K, Na, Ca, and Mg) in the bovine liver, human hair, and tea leaves. Measurement results obtained from LIBS spectra, presented in Table 3, show relatively high limits of detection obtained by LIBS. For higher amounts of K, Na, Ca, and Mg, the measurement results are in reasonable agreement as compared to the reference data (differences at the level of 4.5–27%); the low concentrations (K and Mg in hair, Na in tea leaves, and Ca in the bovine liver) revealed overestimated values. A well-known approach was used to determine LOD values: LOD = 3σ/s, where σ is the background noise and s is the calibration curve slope [13,23]. In our experimental conditions, the LOD values obtained by LIBS for K, Na, Ca, and Mg were2.8 µg/mL, 3.4 µg/mL, 3.1 µg/mL, and 3.3 µg/mL, respectively. Table 3 summarizes the results of our work as compared to CRM values.

LOD values for K and Na in the analyzed solutions found by other authors ranged from 0.006 to 4 mg/L for K, and those for Na ranged from 0.0004 to 7.5 mg/L [24]. Huang et al. developed another method of calibration for the elements in liquid microdroplets [25]. The authors obtained 1.2 ± 0.1 and 0.6 ± 0.1 mg/L as LODs for K and Na, respectively. Our LOD values are within the range reported by other authors. However, they are higher than those typical for AAS, ICP OES, and ICP-MS. For Ca, the LOD values were reported in the range of 0.003–8 mg/ L, and for Mg, they were 0.0004 to 8 mg/L [24]. Other authors [26] calculated the LOD for Ca at 1.9 μg/mL and for Mg at 3.2 μg/mL in aqueous solutions.

After calibration and the comparison of LIBS results with reference ones, a multivariate analysis method—namely factorial analysis—was applied to the LIBS spectra in the entire 200–800 nm spectral interval. The FA conversion results were successful only for Ca and Mg spectra and are shown in Figure 5 and Figure 6.

It can be seen in Figure 5 and Figure 6 that the averaged LIBS spectra (points in the F1–F2 plane), for samples of the adequate element content, carry together over 80% of the input variance. This allows the application of FA in the calibration process. Knowledge of spectral data is not necessary—analysis is carried out on the entire spectra. However, coefficients of determination are lower, which is understandable since, across the whole spectrum, the influence of the matrix composition is stronger than for the selected spectral lines measured in a narrow spectral band.

Unfortunately, the FA calculations failed for K and Na. Although they built corresponding curves as for Ca and Mg, the points representing the LIBS spectra do not follow the changing content of the elements in the samples. This can be attributed to mismatching or large differences between the plasma temperature and the energies of upper levels in K II and Na II, the elements belonging to the first group of the periodic table, namely elements having one valence electron. The energies of upper levels in the strongest atomic transitions in K I and Na I are low (1.6 eV for K and 2.1 eV for Na), while for K II and Na II, they are high (28 eV for K and 40 eV for Na) as compared to the plasma temperature. It is typically approximately 1 eV in the LIBS experiment if the 10–50 mJ pulse laser is used. In the case of Ca and Mg, both atomic and ionic lines of Ca and Mg appeared in the spectrum. The strongest lines belonged to Ca II and Mg II ions. On the other hand, we registered only K I and Na I atomic lines. K II and Na II ionic lines were not registered because of the high energy upper levels of the strongest resonance lines. Consequently, they must be very weak in the ~1 eV plasma and not seen in the spectrum. In contrast to K and Na, resonance lines of Ca II (393 and 396 nm) and Mg II (279 and 280 nm) have their upper levels at approximately3 eV and 4 eV for Ca and Mg, respectively, and were strong and easy to identify in the spectrum. The large difference between the upper level energies of K II and Na II and the plasma temperature, leading to weakly changing spectra, seems to be one of the reasons for the FA calculation’s failure for K and Na samples. Moreover, matrix effects might be stronger for K and Na samples and could be another reason for the FA failure. In Appendix A, a stronger background can be seen in the K and Na spectra. In our future measurements, the experimental conditions will be selected more carefully and separately for each investigated element, taking into account its electron energy structure.

Nevertheless, the discussed results present the reasonable compatibility and power of FA quantitative determination of element concentrations, comparable with spectral methods. In the presented data, the success of FA concerns only Ca and Mg samples—in the case of K and Na, the FA provided somewhat poorer results than those shown in Figure 5 and Figure 6 for Ca and Mg.

## 4. Conclusions

It was shown that, after preparing calibration samples, the LIBS method is suitable for the quantitative determination of K, Na, Ca, and Mg in environmental and biological samples. This was confirmed by the calibration curves presented in the paper, with high coefficients of determination. The greatest contribution to the uncertainty of measurement is represented by the heterogeneity of the sample, the filter paper, and LIBS signal scatter caused mainly by fluctuations of the laser parameters. In the case of K samples, we observed the highest uncertainties of measurement, mainly caused by the much lower signals than for Na, Ca, and Mg samples. The developed method of calibration for the elements in the tested concentration range allows for their quantitative evaluation in samples in the liquid state.

In certain conditions, the statistical factorial analysis can be a useful preliminary approach for the fast, rough determination of the element content in the tested material. An important matter is that we do not need the knowledge of any spectral line data in the spectrum—entire spectra are processed and this is sufficient to perform the calibration. It results also from the presented data that the parameters of LIBS plasma should be matched with the electron energy levels of the investigated elements. This is why the FA was successful for Ca and Mg and not for K and Na.

## Figures and Tables

**Figure 1 materials-15-03736-f001:**
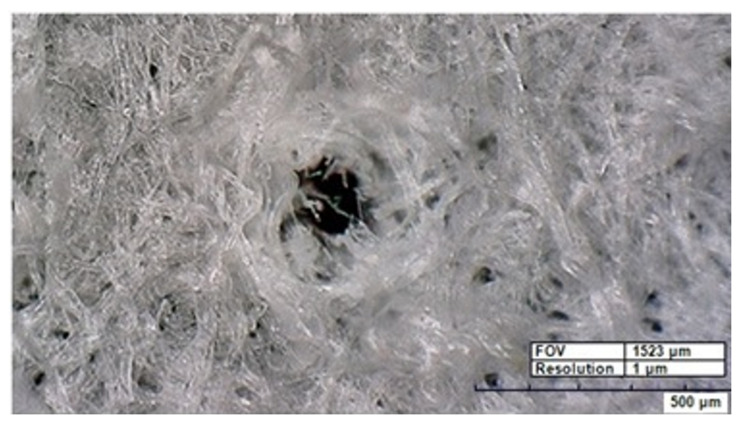
An exemplary view of a filter paper with Na solution after 3 laser shots.

**Figure 2 materials-15-03736-f002:**
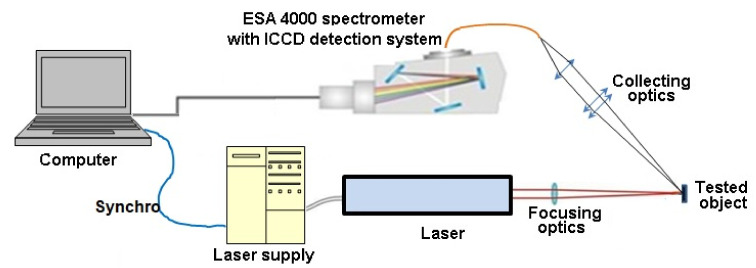
The diagram of the measurement stand.

**Figure 3 materials-15-03736-f003:**
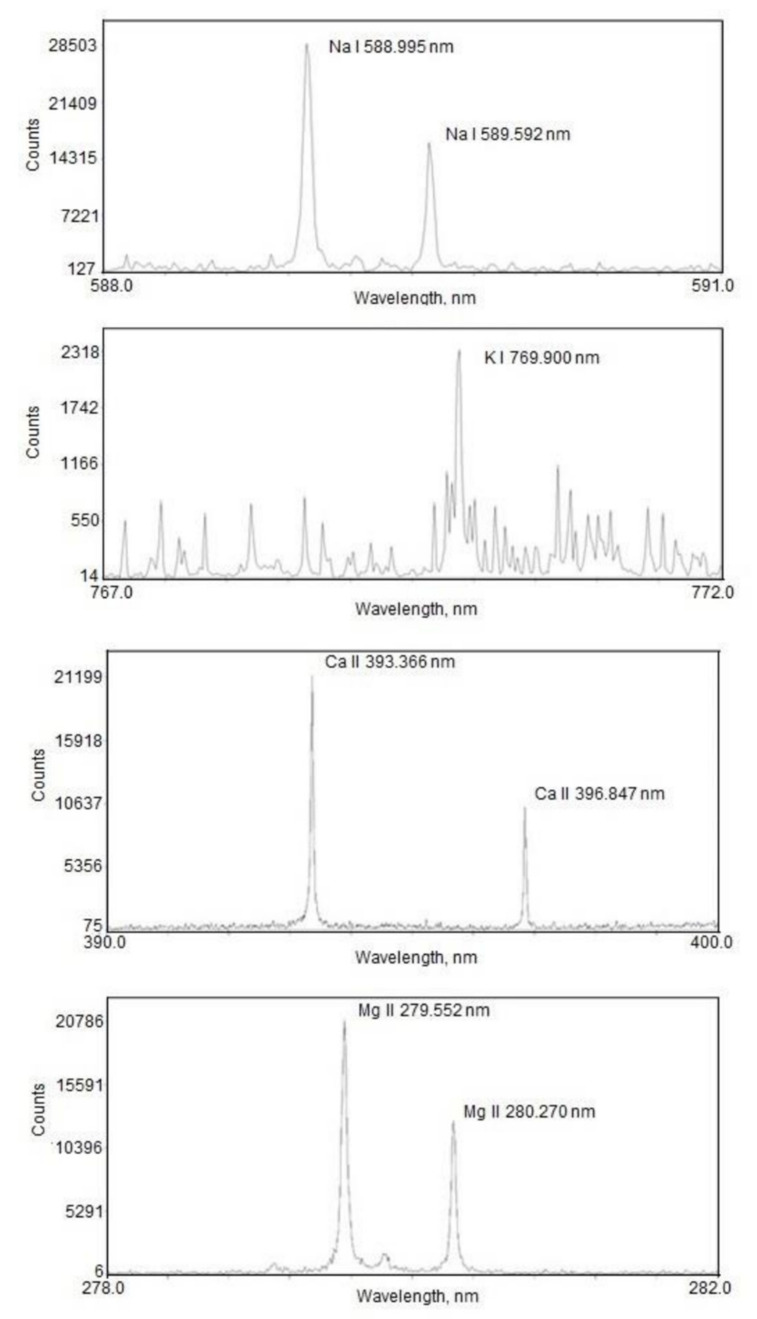
Emission spectra of Na I, K I, Ca II, and Mg II in the regions of interest.

**Figure 4 materials-15-03736-f004:**
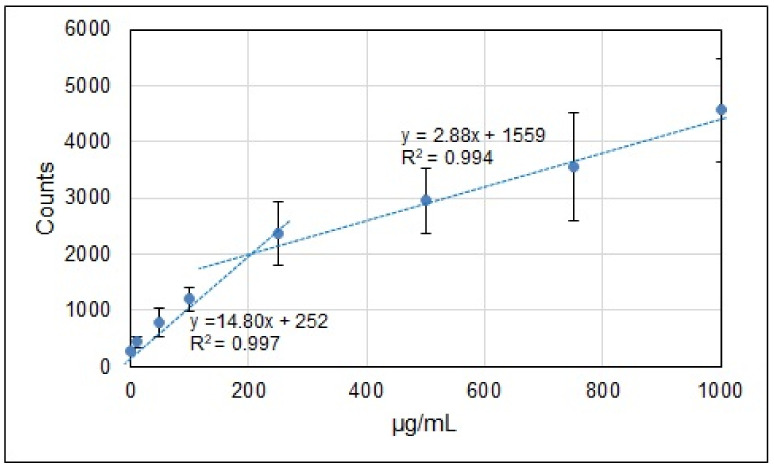
Double-line approximation of the data for K calibration.

**Figure 5 materials-15-03736-f005:**
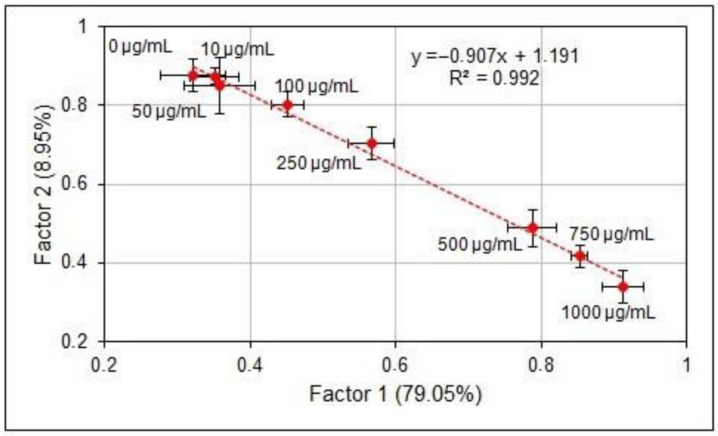
The Ca LIBS spectra in the FA plane.

**Figure 6 materials-15-03736-f006:**
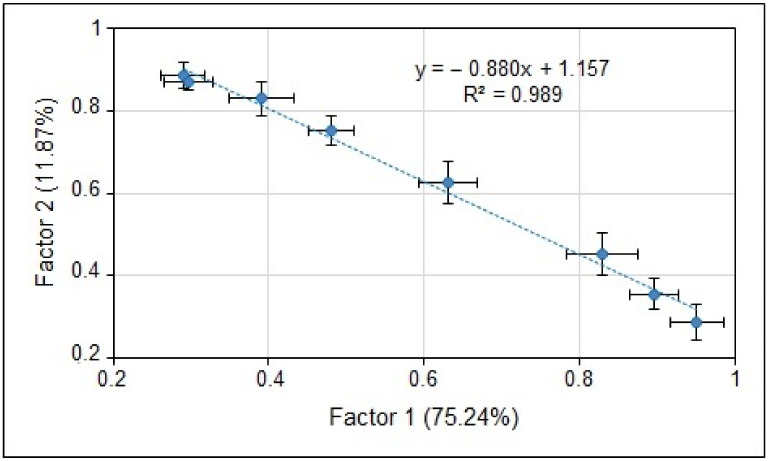
The Mg LIBS spectra in the FA plane.

**Table 1 materials-15-03736-t001:** Examples of the element content.

	Units	K	Na	Ca	Mg	Refs.
Sea water	mg/L	10556	380	400	1262	[1]
Surface water	mg/L	16–63	7–24	58–178 *	58–178 *	[2,3] *
Groundwater	mg/L	6–27	18–110	162–280	25–74	[3]
Intracellular fluids	mmol/L	156	10			[4]
Interstitial fluids	mmol/L	3.8	144.0			[4]
Serum	mEq/L	4	142	5	2	[5]
Red cells	mEq/L	85.78–107.95	3.98–19.67	0.61–1.97	1.6–6.70	[5]
24h urine excretion	mEq/L	25–100	120–220	100–250	4–16	[5]

* it is attributed to ref [3].

**Table 2 materials-15-03736-t002:** Examples of the element content.

	Units	K	Na	Ca	Mg	Refs.
Tap water	mg/100 g	nd ^1^–20.4	0.1–39.1	nd ^1^–10	nd ^1^–6	[7]
Low-mineralized waters	mg/L		1–56	4–15	1–110	[8]
High-mineralized waters	mg/L		900–1419	5–176	4–60	[8]
Black tea infusion	mg/250 mL	42.96	0.23	0.58	2.10	[9]
Coffee infusion	mg/L	887.37–1547.70	24.74–27.81	16.34–25.71	77.15–116.30	[10]

^1^ not detected.

**Table 3 materials-15-03736-t003:** Comparison of the results for K, Na, Ca, and Mg content determined by LIBS method in various CRM samples (μg/mL).

	K			Na			Ca			Mg		
	CRM	LIBS	Δ, %	CRM	LIBS	Δ, %	CRM	LIBS	Δ, %	CRM	LIBS	Δ, %
SRM 1577b Bovine Liver	92.8	104.6 ± 52.0	13	22.6	28.8 ± 6.6	27	1.1	-	-	5.6	5.3	5
NCT-TL-1 Tea Leaves	165.0	157.0 ± 51.0	5	0.2	-	-	56.5	45.2	20	21.8	19.8	9
NCS ZC91002 Human Hair	0.1	-	-	2.8	3.4 ± 0.9	21	10.7	11.35	6	1.0	1.7	70
LOD (μg/mL)	2.8			3.4			3.1			3.3		

## Data Availability

Not applicable.

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
