# Peer review of "Application of Laser-Induced Breakdown Spectroscopy in the Quantitative Analysis of Elements—K, Na, Ca, and Mg in Liquid Solutions"

_materials, 2022, doi:10.3390/ma15103736_

Round 1
Reviewer 1 Report
The article presents the results of using Laser-Induced Breakdown Spectroscopy for measurements of K, Na, Ca, and Mg content in liquid media.
The article is poorly prepared, the English language requires a serious revision (I recommend a service with a native speaker). Lots of typos and inaccuracies. As it stands, the article cannot be recommended for publication. The article is overloaded with unnecessary details (a huge number of numerical values that significantly complicate the perception of the text, etc.). In principle, the results may be of some interest, but the present form of presentation of information does not at all meet the standard of a high-level scientific publication.
More:
What is the novelty of the research? What is the scientific value of the article? This must be clearly stated!
I do not recommend using abbreviations in the title of the article.
"LIBS method" - the word "method" seems superfluous to me. The title "Application of Laser-Induced Breakdown Spectroscopy in the quantitative analysis of elements – K, Na, Ca, and Mg in liquid solutions" can be recommended.
I do not recommend using abbreviations in Abstract. Especially the abbreviation FA - used only once.
It's not very clear what "good conformity" means - it's not a scientific definition. Correct "deviation was 5%" or something like that.
The keywords "potassium; sodium; calcium; magnesium" is a bit of an odd choice for keywords. These are too global concepts!
All sections are number 1!
So,... - not the style of a scientific article.
In general, I recommend additional correction of the English language, especially in terms of technical article standards.
10.556 mg/L, 380 mg/L - in comparable values, the same number of decimal places must be used. In general, why do the authors provide these data? If the authors feel that the values presented are important to the understanding of the article, I recommend presenting this in tabular form. As it stands, the text is difficult to read.
What is EFSA (should the reader know this?)
"methods such as AAS, ICP-OES and ICP-MS" - at least references for describing these methods are needed!
"Analyzed tea (black tea, Ceylon)" - is it important that it is Ceylon? Is there only one type of tea produced in Ceylon? Why is this information needed at all?
The introduction needs to be substantially revised. Now this is a stream of numbers that overload the perception and it is not very clear what they give. I recommend switching to a tabular presentation if the authors are sure of the need for this information.
Why is part of the Introduction with one font, and part - with another?
have been already described by some authors [12-14] - so what did these "some authors" write?
"higher detection limits compared to, e.g., AAS, ICP, ICP-MS, and calibration procedures" - what exactly is this "limit"? Where is it described? What does "e.g." mean? Are there more methods? why are they not listed?
The introduction should be completely changed!
Figure 2 - scale bar is indistinguishable as it stands
"A Hirox 6700 digital ..." - this information should be in the Materials and Methods section.
"y = 38.60x + 1658 and..." - these data are presented in the image - what's the point of repeating them. This is a scientific article, not a laboratory work!
I recommend authors to use tables more often and it makes no sense to repeat numerical values in the text. It is enough to give a reference "the data is presented in Fig... Tab..."
The text should contain only conclusions from the analysis of the results.
References do not meet the requirements of the publisher.
Reviewer 2 Report
This paper reports the use of the LIBS technique to determine the concentration of Na, K, Ca, and Mg in liquid solutions dried on a filter paper disk (presumably) a few cm2 in area. LIBS calibration obtained from prepared samples was tested with certified biological samples. This work is a direct extension of previous work (Ref. 17) by the same authors dealing with the quantification of Al and Si in liquid solutions using the same method.
Overall, the paper is interesting and the results are honestly presented and discussed. Fundamentally, the proposed technique raises questions because the advantage of LIBS over more accurate conventional methods lies in its ability to provide much faster responses while the proposed technique involves drying samples. This point would need to be discussed.
Beyond this concern, one may wonder why the authors did not use an internal normalization in their calibration curves, presented as counts vs concentration, to lessen the usual problems of LIBS, namely variations in the amount of ablated material and sample heterogeneity. Normalization to the integral of a portion or the entire spectrum are the most common options. In addition, PCA (and probably Factorial Analysis) generally benefits from appropriate normalization. For calibration curves, it should be mentioned whether it is the net intensity of the lines (peak minus background) that has been taken into account as this should be done.
In conclusion, I found this paper interesting and recommend the authors to consider the points raised above before submitting an updated version.
Reviewer 3 Report
Dear authors,
I read the paper entitled (Application of LIBS method in the quantitative analysis of elements—K, Na, Ca, and Mg in liquid solutions). Here is my report:
The authors presented an experimental LIBS for measurements of K, Na, Ca, and Mg content in liquid media. The LIBS-based procedure in conjunction with a Chemometric method (FA) to measure the cited above elements has been presented.
The tools used and the presentation of results lack a useful and sounds interpretation. It needs a major revision along with the questions that should be answered before its publication. I strongly do not recommend it for publication without the required improvement.
The paper could not be presented as it is. I recommend a professional editor revise the manuscript after it is written more concisely.
The introduction should be much shorter, avoiding providing a lot of numbers with various units, which is confusing.
Some specific comments:
Page 2, line 60: The reference must be double-checked for spelling errors.
Page 2, line 61: LIBS advantages should be described and compared to traditional methods.
Figures: error bars must be added.
Page 2: Materials and experimental section: For calibration, authors used filter paper as a substrate; However to verify the calibration (test step), they used liquid samples: The authors should clarify the measurement procedures. Spectra should also be presented. I’m suspicious about a strong matrix effects induced by using a substrate.
Page 3; line 99: please precise that the 4 ns is the pulse width. Also, the detection parameters should be clarified.
Page3; line 112: data jitter: what does this mean? Repeatability?
Page 5: Figure 3 showed potassium line at 769.9 nm. There is another line K line at 766 nm, which is more sensitive. Why the authors did not show and use the K line 766nm for the calibration since it is available in the spectra acquired by the ESA spectrometer?
Page 6: Figure 4: How do the authors explain the double line approximation? Is it due to the self-absorption of the potassium line?
Page 7, line 195: Factorial Analysis must be described.
Summary: elements given in this paper are important to be published and but they lack of clarity, some explanation and good arguments for the approaches used. I strongly do not recommend it for publication without the required improvements.
Round 2
Reviewer 1 Report
In general, the authors have done a great job in order to improve the quality of the manuscript. The reviewer's recommendations were generally taken into account. I believe that the menuscript can be recommended for publication, but I draw the attention of the authors to:- Tables should be submitted in the format recommended by the editors. - "The novelty and advantage of the paper..." is not quite the correct form for the Conclusion. In principle, it is understood that everything presented in the Conclusion is novelty. - As I noted in the first review, you cannot compare numbers with different numbers of decimal places. 3.98-19.67 and 7-24, 18-109.7 etc. - The same applies to Table 2. 887.37-1540.70 and 24.736-
27.810 - what accuracy is exactly needed?
Reviewer 3 Report
The authors improved the quality of the manuscript. The reviewer's recommendations were generally taken into account. I believe that the manuscript can be accepted for publication after minor revision:
- Tables should be submitted in the format recommended by the editors.
- Page 3: Please replace “internal amplification” by “Camera Gain”
- Decimal numbers should be fixed.
